# The Role of Pretreatment Serum Interleukin 6 in Predicting Short-Term Mortality in Patients with Advanced Pancreatic Cancer

**DOI:** 10.3390/biomedicines12040903

**Published:** 2024-04-18

**Authors:** Se Jun Park, Ju Yeon Park, Kabsoo Shin, Tae Ho Hong, Younghoon Kim, In-Ho Kim, MyungAh Lee

**Affiliations:** 1Division of Medical Oncology, Department of Internal Medicine, The Catholic University of Korea, Seoul St. Mary’s Hospital, Seoul 06591, Republic of Korea; psj6936@naver.com (S.J.P.); agx002@naver.com (K.S.); 2Cancer Research Institute, College of Medicine, The Catholic University of Korea, Seoul 06591, Republic of Korea; pke1001@hanmail.net; 3Department of General Surgery, The Catholic University of Korea, Seoul St. Mary’s Hospital, Seoul 06591, Republic of Korea; gshth@catholic.ac.kr; 4Department of Pathology, College of Medicine, The Catholic University of Korea, Seoul St. Mary’s Hospital, Seoul 06591, Republic of Korea; sellar1@snu.ac.kr

**Keywords:** pancreatic cancer, interleukin-6, prognosis, inflammation

## Abstract

Pancreatic ductal adenocarcinoma (PDAC) is notorious for its aggressive progression and dismal survival rates, with this study highlighting elevated interleukin 6 (IL-6) levels in patients as a key marker of increased disease severity and a potential prognostic indicator. Analyzing pre-treatment serum from 77 advanced PDAC patients via ELISA, the research determined optimal cutoff values for IL-6 and the IL-6:sIL-6Rα ratio using receiver operating characteristic curve analysis, which then facilitated the division of patients into low and high IL-6 groups, showing significantly different survival outcomes. Notably, high IL-6 levels correlated with adverse features such as poorly differentiated histology, higher tumor burden, and low albumin levels, indicating a stronger likelihood of poorer prognosis. With a median follow-up of 9.28 months, patients with lower IL-6 levels experienced markedly better median overall survival and progression-free survival than those with higher levels, underscoring IL-6’s role in predicting disease prognosis. Multivariate analysis further confirmed IL-6 levels, alongside older age, and elevated neutrophil-to-lymphocyte ratio, as predictors of worse outcomes, suggesting that IL-6 could be a critical biomarker for tailoring treatment strategies in advanced PDAC, warranting further investigation into its role in systemic inflammation and the tumor microenvironment.

## 1. Introduction

Most patients with pancreatic ductal adenocarcinoma (PDAC) present an advanced stage at diagnosis that is not amenable to curative surgery [1]. In 2020, the 5-year survival rate for PDAC neared 10% for the first time, a slight increase from 5.26% in 2000 [1]. This modest improvement in survival rates is chiefly attributed to multi-agent cytotoxic treatment regimens, however, the efficacy of cytotoxic chemotherapy remains unsatisfactory [2,3,4].

According to several studies previously published, systemic inflammation is thought to be associated with tumorigenesis, proliferation, invasion, angiogenesis, and metastasis of pancreatic cancer [5,6]. In patients with pancreatic cancer, markers representing systemic inflammation, such as neutrophil-to-lymphocyte ratio (NLR), platelet-to-lymphocyte ratio, and C-reactive protein, have been identified as being associated with prognosis [7,8,9]. Additionally, a recently published meta-analysis has shown the potential for predicting prognosis through inflammation indexes, which are quantified using markers that represent systemic inflammation [10].

Cytokines play a crucial role in regulating various biological functions such as cell proliferation and differentiation, immune responses, inflammation, and metabolic processes [11]. In the context of pancreatic cancer, an elevated expression of both pro-inflammatory and anti-inflammatory cytokines has been observed in serum samples. Interleukin-6 (IL-6), a pro-inflammatory cytokine, facilitates cancer development by activating the Janus kinase 2 (JAK2)–signal transducer and activator of transcription 3 (STAT3) pathways [12]. This leads to enhanced tumor vascularization, which in turn promotes the migration and metastasis of cancer cells [13]. Additionally, in patients with advanced PDAC, elevated pretreatment levels of IL-6 are associated with poor survival outcomes and resistance to chemotherapy [12]. Elevated concentrations of interleukin-1β (IL-1β) are correlated with increased tumor aggressiveness and have been implicated in the facilitation of metastatic processes [14,15]. Interleukin-10 (IL-10), a potent immunosuppressant, has been found to contribute to immune evasion and is associated with poor prognosis [16]. However, the relationship between serum proinflammatory cytokines and the effectiveness of systemic chemotherapy, along with survival outcomes in patients with advanced pancreatic cancer, remains largely unexplored. Investigating the association between pretreatment proinflammatory cytokines and short-term mortality or chemotherapy resistance in these patients could prevent unnecessary treatment in those predicted not to respond. Additionally, elucidating the relationship between IL-6 and systemic inflammatory biomarkers may help characterize the features of pancreatic cancer that are commonly known to be unresponsive to immunotherapy, potentially leading to new therapeutic strategies.

This study explored the correlation between pretreatment serum IL-6 levels, encompassing pro-inflammatory cytokine levels, and survival outcomes in advanced pancreatic cancer patients, while also assessing how these biomarkers influence the efficacy of systemic chemotherapy. Additionally, we examined the correlation between IL-6 and the systemic inflammatory marker, NLR, by analyzing the proportion and absolute count of neutrophils and lymphocytes according to IL-6 levels.

## 2. Materials and Methods

### 2.1. Patients and Samples

Between 2017 and 2021, 77 patients with advanced PDAC were enrolled from the Catholic University of Korea, Seoul St. Mary’s Hospital. Eligibility criteria for patients with available serum samples at the time of advanced disease diagnosis included: (1) histological confirmation of PDAC; (2) presence of locally advanced or metastatic disease; (3) survival verification at the time of data collection. Prior to initiating first-line systemic chemotherapy in a palliative care setting, we collected peripheral blood samples from these patients.

### 2.2. Enzyme-Linked Immunosorbent Assay (ELISA)

Peripheral blood samples were obtained from patients prior to the initiation of any systemic chemotherapy, aimed for palliative care. Upon receipt in the laboratory, these samples were promptly centrifuged at 2500× *g* for 10 min and subsequently aliquoted into volumes of 2–4 mL for storage at −80 °C. The quantification of cytokines in the plasma utilized ELISA kits designed for IL-6 (#DY206-05), leukemia inhibitory factor (LIF) (#DYF00B), IL-1β (#DY201-05), IL-10 (#DY217B-05), granzyme B (#DY2906-05), and interferon-gamma (#DY285B-05), all sourced from R&D Systems, Minneapolis, MN, USA, and were employed following the manufacturer’s specified protocols. Additionally, levels of soluble IL-6 receptor α (sIL-6Rα) protein were measured employing DuoSet ELISA kits for each respective protein (#DY227, R&D Systems, Minneapolis, MN, USA), adhering to the provided instructions.

### 2.3. Defining a Cohort with High Serum IL-6 Levels

Serum IL-6 levels were found to be higher in patients with metastatic PDAC than in those with locally advanced disease. Furthermore, a correlation was observed between elevated serum IL-6 levels and reduced overall survival [17]. However, due to the limited scope of studies involving small sample sizes, an optimal cutoff for IL-6 and the IL-6:sIL-6Rα ratio that correlates with survival outcomes in metastatic PDAC patients has yet to be determined. As a result, a specific cutoff value of IL-6 for prognostication in PDAC patients remains undefined. To address this, we employed receiver operating characteristic (ROC) curve analysis to define an optimal prognostic cutoff value for IL-6 and the IL-6:sIL-6Rα ratio in predicting survival six months after enrollment. This allowed us to classify patients into two cohorts—those with high versus low levels of IL-6 and the IL-6:sIL-6Rα ratio—based on the ideal cutoff value.

### 2.4. Flow Cytometry for Detection of T Cells

Using Ficoll-Paque PLUS (#GE17-1440-02, Sigma-Aldrich, Burlington, MA, USA) for density gradient centrifugation, peripheral blood mononuclear cells were separated and then preserved in liquid nitrogen for subsequent analysis. Once thawed, these cells were treated with anti-CD3 (#317314), anti-CD4 (#357416), anti-CD45 (#368510), and anti-CD8 (#300910) antibodies from BioLegend, CA, USA, incubating for 20 min at 4 °C in darkness. Following staining, flow cytometry was employed for cell analysis using a BD FACSCanto II instrument (BD Biosciences, San Jose, CA, USA), with FlowJo software version 10.8 (Tree Star, Ashland, OR, USA) utilized for data analysis.

### 2.5. Statistical Analysis

Descriptive statistics were presented as medians with ranges or interquartile ranges (IQR), and proportions. To evaluate the predictive accuracy of clinical factors for survival outcomes, a time-dependent ROC curve analysis was conducted. Youden’s index was employed to determine the optimal cutoff values for these clinical factors. The relationship between clinicopathological characteristics and serum IL-6 levels, as well as differences between groups, were analyzed using chi-squared or Fisher’s exact tests for categorical variables and the unpaired *t*-test with Welch’s correction for continuous variables. The correlation between IL-6 levels and the IL-6:sIL-6Rα ratio was assessed using Pearson’s correlation coefficient. OS was defined as the duration in months from the advanced disease diagnosis until death from any cause or the last follow-up. Progression-free survival (PFS) was defined as the time between the start of chemotherapy and either documented disease progression or death. Survival estimates were generated via the Kaplan–Meier method, with the log-rank test applied to compare survival differences using a two-tailed approach. Cox proportional hazards regression models identified clinical factors impacting survival, estimating hazard ratios (HR) and 95% confidence intervals (CIs). Statistical significance was attributed to results with two-sided *p*-values below 0.05. The statistical analyses were executed using SPSS for Windows (version 24.0; IBM SPSS Inc., Armonk, NY, USA) and GraphPad Prism version 10.2.0 (GraphPad Software Inc., San Diego, CA, USA).

## 3. Results

### 3.1. Clinicopathological Characteristics of the Study Cohort

Table 1 outlines the clinicopathologic features of 77 patients diagnosed with advanced PDAC. The median age was 67 years, with the majority (80.5%) having an Eastern Cooperative Oncology Group (ECOG) score of 0–1. Regarding the location of the primary tumor, 31 patients (40.2%) had it in the pancreatic head, while 46 patients (59.8%) had it in the body and tail of the pancreas. Among them, 69 patients (89.6%) presented with systemic disease, 8 patients (10.4%) had locally advanced disease without distant metastasis, 62 patients (80.5%) were already in an advanced stage at diagnosis, and 15 patients (19.5%) experienced recurrence after surgery. More than half, 42 patients (54.5%), had well or moderately differentiated tumors, and the majority, 65 patients (84.1%) showed metastasis in fewer than two organs. Serum carbohydrate antigen 19-9 (CA 19-9) levels were more than 59 times higher than normal in 54 patients (70.1%), and 51 patients (66.2%) maintained normal albumin levels at baseline.

### 3.2. IL-6 Levels in Patients with Advanced Pancreatic Cancer

IL-6 was undetectable or below the lower detection limit in 10 of the 77 patients (13.0%). The median IL-6 concentration was found to be 5.13 pg/mL (IQR, 1.65–10.5 pg/mL), with the median IL-6-to-sIL-6Rα ratio (IL-6:sIL-6Rα) being 2.45 (IQR, 0.77–5.01 pg/ng). There was a strong association between the IL-6:sIL-6Rα to the corresponding levels of IL-6 (*Rs* = 0.984; *p* < 0.0001; Appendix A). The serum levels of IL-1β and IL-10 did not show a correlation with IL-6 levels. However, serum LIF demonstrated a correlation with IL-6 (*Rs* = 0.112; *p* = 0.0029; Appendix A), although it was only measured in 19 subjects (24.7%). Notably, patients exhibiting a higher histological grade (*p* = 0.004), greater number of metastatic sites (*p* = 0.025), and lower albumin levels (*p* = 0.006) presented with significantly higher IL-6 levels than those with lower histological grade, fewer metastatic sites, and higher albumin levels (Table 1). Among patients with metastases, those with liver metastases exhibited higher IL-6 levels compared to those without liver involvement (*p* = 0.034). Moreover, no correlation was found between IL-6 levels and other clinical or pathological features, including disease status or initially elevated CA 19-9 levels.

### 3.3. IL-6 Levels and Survival Outcomes

As of the final analysis on 31 December 2023, 70 patients (90.9%) had died, with 92.9% (65 patients) succumbing to disease progression and the remaining 7.1% (5 patients) to treatment-related adverse events. The median follow-up was 9.28 months (95% CI, 6.10–12.6). The median OS for the entire cohort was 9.28 months (95% CI, 6.24–12.3), and OS rates at 6 months, 9 months, and 1 year were 62.3%, 50.6%, and 39.0%, respectively. Using the ROC curve, an optimal IL-6 cutoff value was determined at 6.03 pg/mL for predicting 6-month survival, achieving a sensitivity of 71.4% and a specificity of 83.7% (Area Under the Curve [AUC] = 0.738, 95% CI, 0.62–0.86, *p* < 0.001, Appendix A). Similarly, an optimal IL-6:sIL-6Rα ratio cutoff was set at 2.58 pg/ng, with an AUC of 0.744 (95% CI, 0.62–0.87), a sensitivity of 78.6%, and a specificity of 75.5% (*p* < 0.001, Appendix A).

Based on these cutoff values, patients were categorized into low and high IL-6 groups (49 and 28 patients, respectively), as well as into low and high IL-6:sIL-6Rα ratio groups (43 and 34 patients, respectively). The low IL-6 group showed a median OS of 13.3 months (95% CI, 10.5–16.0) compared to 3.63 months (95% CI, 2.20–5.01) in the high IL-6 group (HR = 0.33; 95% CI, 0.18–0.60; *p* < 0.0001; Figure 1A). For the IL-6:sIL-6Rα ratio, the low group had a median OS of 13.4 months (95% CI, 11.0–15.9) versus 3.64 months (95% CI, 1.72–5.56) in the high group (HR = 0.32; 95% CI, 0.18–0.56; *p* < 0.0001; Figure 1B). In the subset receiving systemic treatment (65 patients), those with low IL-6 levels or low IL-6:sIL-6Rα ratios experienced significantly longer survival times—13.5 months (95% CI, 11.0–15.9) and 13.4 months (95% CI, 11.5–15.3), respectively—compared to 4.62 months (95% CI, 1.83–7.42) (HR = 0.33; 95% CI, 0.16–0.67; *p* < 0.0001; Figure 1C) and 5.61 months (95% CI, 3.20–8.01) (HR = 0.34; 95% CI, 0.18–0.65; *p* < 0.0001; Figure 1D) for those in the high groups.

### 3.4. IL-6 Levels and Effectiveness of Systemic Treatment

Among patients who received palliative systemic treatment, those with the high IL-6 group showed a tendency to receive more gemcitabine-based treatments as first-line chemotherapy compared to FOLFIRINOX, than those with the low IL-6 group (*p* = 0.013, Table 2). Significant differences in median PFS were noted, with the low IL-6 group showing 9.12 months (95% CI, 6.71–11.5) versus only 2.07 months (95% CI, 1.53–2.61) in the high IL-6 group (HR = 0.26; 95% CI, 0.12–0.58; *p* < 0.0001; Figure 2A). Similarly, when comparing IL-6:sIL-6Rα ratios, the low ratio group exhibited a median PFS of 9.12 months (95% CI, 6.83–11.4), significantly longer than the 3.15 months (95% CI, 0.55–5.75) observed in the high ratio group (HR = 0.32; 95% CI, 0.17–0.63; *p* < 0.0001; Figure 2B). While the response rates between the groups did not differ significantly, the analysis revealed a significantly enhanced disease control rate in the cohort with lower IL-6 levels (*p* = 0.001). Furthermore, the PFS rate at six months was substantially higher in the low IL-6 group, at 64.9%, compared to 10.7% in the high IL-6 group (Table 2). Additionally, efficacy outcomes for chemotherapy showed similar results in groups divided based on IL-6:sIL-6Rα ratios (Appendix A).

### 3.5. Multivariate Analysis for OS

The results of the univariate and multivariate analyses for OS are presented in Table 3, with subgroups based on the clinical features and IL-6 levels. Higher IL-6 levels were associated with higher mortality in the univariate analysis. A significant relationship was also observed between IL-6 levels and OS in the multivariate analysis (HR = 2.31; 95% CI, 1.27–4.20; *p* = 0.006; Table 3). Additionally, in the multivariate analysis, older age (age ≥ 65 vs. <65; HR = 1.89; 95% CI, 1.02–3.47; *p* = 0.024) and a higher baseline NLR (NLR ≥ 3.5 vs. <3.5; HR = 3.26; 95% CI, 1.77–6.00; *p* < 0.001) were significantly associated with worse OS. Furthermore, patients who received systemic chemotherapy had longer OS (HR = 0.29; 95% CI, 0.14–0.61; *p* = 0.001) compared to those who did not receive it (Table 3).

### 3.6. Relationship between IL-6 and Neutrophils, Lymphocytes in Patients with PDAC

A significantly positive correlation was detected between the levels of serum IL-6 and baseline NLR (*Rs* = 0.150; *p* = 0.0005; Figure 3A). Notably, the NLR values differed significantly between the IL-6 high and low groups, with the former demonstrating considerably elevated NLR levels (*p* = 0.0004; Figure 3B). Examining the correlation between IL-6 levels and the proportions of neutrophils and lymphocytes revealed that the IL-6 high group was associated with a significantly increased proportion of neutrophils (*p* = 0.0003; Figure 3C) and a reduced proportion of lymphocytes compared to the IL-6 low group (*p* < 0.0001; Figure 3D). Additionally, the analysis showed a statistically significant rise in the absolute neutrophil count in the IL-6 high group when compared to the low group (*p* = 0.0001; Figure 3E), whereas the absolute lymphocyte count was observed to be higher in the IL-6 low group than in the high group (*p* = 0.0246; Figure 3F). Furthermore, in comparing the subpopulations of circulating T cells, the proportions of CD3 + CD45 + CD4 + T cells (*p* = 0.909; Figure 3G) and CD3 + CD45 + CD8 + T cells (*p* = 0.845; Figure 3H) showed no significant variance between the groups categorized by IL-6 levels.

## 4. Discussion

Previous reports indicate that systemic inflammation and elevated IL-6 levels are associated with an unfavorable prognosis in patients with pancreatic cancer [18,19]. However, the relationship between baseline IL-6 levels in treatment-naïve patients with advanced pancreatic cancer and their subsequent prognosis, as well as the effectiveness of systemic chemotherapy and its connection with systemic inflammatory markers, remains largely unexplored. To our knowledge, this study is the first to demonstrate an association between serum IL-6 levels and the IL-6:sIL-6Rα ratio with short-term mortality and the efficacy of systemic chemotherapy. It also establishes their correlation with the NLR, a major systemic inflammatory marker, in patients with advanced pancreatic cancer.

Investigations into the association between IL-6 and sIL-6Rα concentrations revealed no significant correlation. Moreover, the ratio of IL-6 to sIL-6Rα, which potentially reflects the inflammatory condition’s severity [20], was found to be predominantly influenced by IL-6 levels, rather than variations in sIL-6Rα levels. Upon secretion into the bloodstream, IL-6 promptly forms a complex with sIL-6R and soluble glycoprotein 130 (sgp130), facilitating pro-inflammatory trans-signaling [21]. Given the substantially higher concentration of sIL-6R compared to IL-6 in circulation, it is posited that the intensity of the inflammatory response is more directly attributable to IL-6 than to sIL-6R levels [21]. This inference remains consistent even when accounting for the IL-6:sIL-6R ratio. Consequently, the magnitude of serum IL-6 levels and their capacity to form the IL-6:sIL-6R:gp130 complex, thereby initiating trans-signaling, is deemed to significantly influence the degree of systemic inflammation.

LIF, a member of the IL-6 cytokine family, is associated with treatment response in PDAC, and LIF blockade is gaining attention as a target for new therapies due to its ability to augment the efficacy of chemotherapy [22]. In this study, LIF levels showed a significant correlation with IL-6, but the interpretation of results is limited by the considerable number of subjects in whom LIF was not detected.

Regarding the association between IL-6 and clinical features, interestingly, IL-6 levels were found to be higher in subjects with poorly differentiated histology, higher tumor burden, lower albumin levels, and liver metastasis. These findings suggest that patients with aggressive tumor biology or substantial tumor load may exhibit a pronounced systemic inflammatory environment, represented by elevated IL-6 levels. Consistent with prior research linking elevated serum IL-6 to cachexia [23,24], our findings also indicate that subjects with decreased albumin levels present with significantly elevated IL-6 concentrations.

In cohorts stratified by predetermined cutoff values for IL-6 and the IL-6:sIL-6Rα ratio, significant predictors of 6-month mortality, a discernible distinction in survival outcomes was observed correlating with these biomarkers. Elevated pretreatment levels of IL-6 were associated with accelerated disease progression and notably reduced survival durations, aligning with the extant literature on metastatic PDAC that correlates increased serum IL-6 with adverse prognoses [25,26]. Moreover, the analysis focused solely on treated patients showed a correlation between high IL-6 levels and both poor OS and reduced PFS, suggesting a potential link between IL-6 and resistance to chemotherapy. These findings align with prior research indicating that patients with elevated IL-6 levels experienced less favorable PFS outcomes when treated with gemcitabine monotherapy or the FOLFIRINOX regimen [27,28]. Unlike previous studies, which either overlooked other clinical features or involved relatively small sample sizes, our study included a larger patient cohort and incorporated a variety of clinicopathological factors in the survival analysis, making our findings more significant.

Regarding resistance to chemotherapy induced by IL-6, prior studies indicate that augmented IL-6 levels, which precipitate the activation of the JAK2-STAT3 pathway, are implicated in chemotherapy resistance [29,30]. Additionally, pro-inflammatory cytokines, including IL-6 within the PDAC tumor microenvironment (TME), are known to facilitate the evasion of anti-tumor immune responses and contribute to chemotherapy resistance [31]. Therefore, inhibiting the IL-6 pathway to suppress systemic inflammation or mitigate the impact of IL-6 on tumor and immune cells within the TME remains an attractive potential therapeutic target.

In multivariate analysis for OS, elevated IL-6 levels were identified as predictors of worse survival outcomes, along with high NLR and older age. Given that IL-6 adversely impacts the prognosis in patients with pancreatic cancer, blocking the IL-6 signaling pathway has been proposed as a potentially effective strategy for those with advanced PDAC [32]. However, tocilizumab, a humanized monoclonal antibody against the IL-6R, in combination with cytotoxic chemotherapy in patients with advanced pancreatic cancer did not demonstrate an improvement in survival outcomes [33]. This inefficacy may stem from the higher concentrations of sIL-6R compared to IL-6 itself, hinting that targeting sIL-6R might not yield significant benefits. Moreover, the rapid formation of a heterodimer complex between plasma-secreted IL-6, sIL-6R, and sgp130 raises the possibility of therapeutic targeting. Using an antibody like the sgp130Fc protein to target the IL-6/sIL-6R complex could offer a therapeutic advantage [21].

As expected, the systemic inflammatory marker NLR was found to have a strong correlation with the levels of serum IL-6. A previous meta-analysis has indicated that an elevated NLR is associated with a poorer prognosis in patients with pancreatic cancer [34]. Our findings indicate that high IL-6 levels, correlating with an increased proportion of neutrophils and a decreased proportion of lymphocytes, lead to a rise in the NLR. Regarding absolute white blood cell counts, individuals with elevated IL-6 levels exhibited notably higher absolute neutrophil counts, while absolute lymphocyte counts were maintained at a relatively constant level between the two groups. Furthermore, no substantial differences were observed in the proportions of CD4+ and CD8+ lymphocytes between the groups. These findings indicate that the adverse prognosis linked to a higher NLR is more significantly influenced by severe systemic inflammation from increased neutrophils, rather than by the immunosuppressive effects stemming from a reduction in lymphocytes. Therefore, treatment strategies for pancreatic cancer that focus on reducing systemic inflammation and preventing neutrophil infiltration into the tumor might be more beneficial than targeting immunosuppressive mechanisms alone.

Several limitations of our study should be noted. Firstly, this research was conducted as a retrospective study at a single institution and involved a relatively small sample size, both of which could act as confounding factors. Secondly, the relationship between treatment efficacy and IL-6 levels was not thoroughly assessed, as measurements of IL-6 were not taken at multiple points after treatment. Additionally, IL-6 levels can increase due to bacterial infections, which raises the possibility that elevated mortality might have resulted from infections rather than tumor progression. Furthermore, the absence of evaluations for IL-6 and tumor-infiltrating T cells in the TME indicates that further research is necessary to explore the immunosuppressive effects exerted by IL-6 in a paracrine manner within the tumor tissue. Lastly, the potential activation of downstream pathways in the tumor tissue, possibly due to IL-6 trans-signaling, needs to be assessed.

In conclusion, pretreatment IL-6 levels and the IL-6:sIL-6Rα ratio were associated with worse survival outcomes and chemotherapy resistance, with this association consistent in multivariate analysis. These insights could enhance the management of patients with advanced PDAC by identifying those most likely to benefit from systemic treatment. IL-6 was closely linked to exacerbated systemic inflammation, primarily driven by an increase in neutrophils. Further research is essential to explore how circulating IL-6 levels relate to its presence in the TME, understand its immunosuppressive role, and examine the correlation between increased circulating and tumor-infiltrating neutrophils in advanced pancreatic cancer.

## Figures and Tables

**Figure 1 biomedicines-12-00903-f001:**
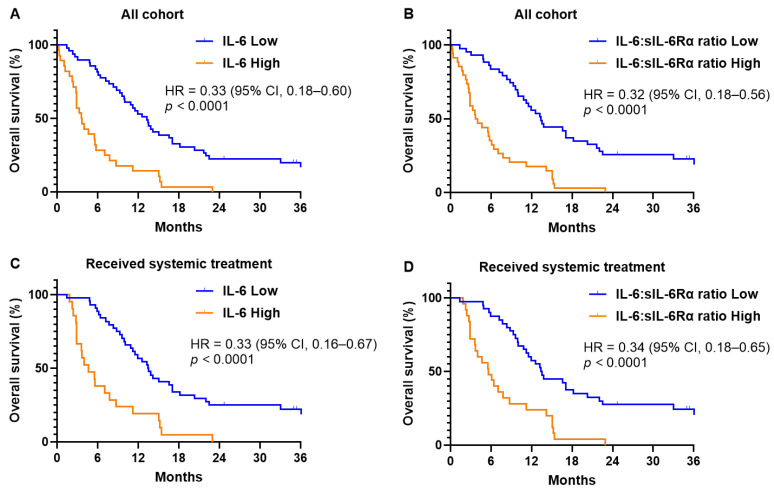
Kaplan–Meier survival curves depicting overall survival based on serum interleukin-6 levels and the ratio of interleukin-6 to soluble interleukin-6 receptor. Panels (**A**,**B**) show survival across all study participants (*n* = 77), while panels (**C**,**D**) focus on patients who underwent systemic treatment (*n* = 65).

**Figure 2 biomedicines-12-00903-f002:**
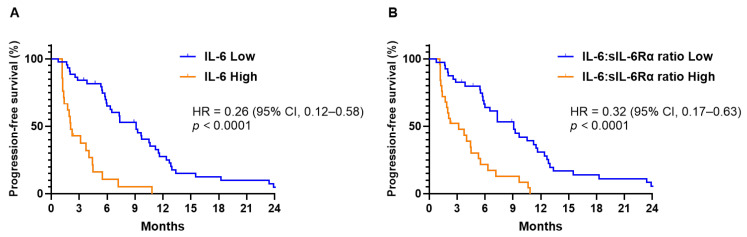
Progression-free survival (PFS) in patients with advanced pancreatic cancer who received palliative systemic chemotherapy, (**A**) categorized by serum interleukin-6 levels and (**B**) differentiated by the ratio of interleukin-6 to its soluble receptor.

**Figure 3 biomedicines-12-00903-f003:**
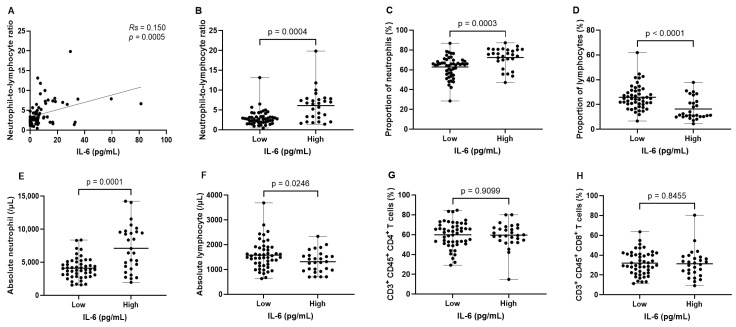
The relationship between interleukin-6 levels and the proportion and absolute count of neutrophils and lymphocytes in patients with advanced pancreatic cancer. (**A**) Correlation between serum IL-6 and neutrophil-to-lymphocyte ratio (NLR), (**B**) difference in NLR values between IL-6 high and low groups, (**C**) differences in neutrophil proportion, (**D**) lymphocyte proportion, (**E**) absolute neutrophil count, and (**F**) absolute lymphocyte count between the two groups according to IL-6 levels, differences in (**G**) CD4+ and (**H**) CD8+ T cell subpopulations between the two groups based on IL-6 levels.

**Table 1 biomedicines-12-00903-t001:** Clinicopathological characteristics and serum interleukin-6 level of patients with advanced pancreatic ductal adenocarcinoma.

Variables	Total (*n* = 77)	IL-6 (pg/mL, Median, IQR)	*p* Value
Age, median (range)	67 (39–86)		0.738
<65-year, *n* (%)	28 (36.4)	5.15 (1.41–16.1)	
≥65-year, *n* (%)	49 (63.6)	4.71 (1.77–9.95)	
Gender, *n* (%)			0.636
Male	44 (57.1)	4.92 (1.41–7.89)	
Female	33 (42.9)	5.20 (1.77–12.0)	
ECOG performance status, *n* (%)			0.409
0–1	62 (80.5)	4.91 (1.69–10.9)	
2	15 (19.5)	6.13 (1.23–10.1)	
Tumor location, *n* (%)			0.229
Head	31 (40.2)	4.70 (1.36–6.35)	
Body/tail	46 (59.8)	5.18 (1.73–11.2)	
Disease status, *n* (%)			0.998
Locally advanced	8 (10.4)	5.42 (0.58–14.0)	
Metastatic	69 (89.6)	5.11 (1.65–10.5)	
Previous tumor resection, *n* (%)			0.688
No (initially advanced)	62 (80.5)	5.18 (1.69–10.9)	
Yes (recurrent disease)	15 (19.5)	5.11 (0.09–6.35)	
Histologic grading, *n* (%)			
Grade 1/2	42 (54.5)	4.04 (0.77–6.47)	0.004
Grade 3	17 (22.1)	13.2 (5.52–17.6)	
Not available	18 (23.4)		
Number of metastatic organ sites *			0.025
1–2	58 (84.1)	4.21 (1.13–6.50)	
≥3	11 (15.9)	16.6 (8.0–29.5)	
Baseline CA19-9 level, *n* (%)			
<59 × ULN (U/mL)	54 (70.1)	4.04 (0.63–6.03)	0.394
≥59 × ULN (U/mL)	22 (28.6)	7.21 (5.05–16.9)	
Unknown	1 (1.3)		
Baseline albumin, *n* (%)			0.006
≥3.5 g/dL	53 (68.8)	3.94 (0.73–5.88)	
<3.5 g/dL	24 (31.2)	11.0 (4.83–26.4)	
Site of metastatic disease *, *n* (%)			
Liver	47 (68.1)	5.55 (1.91–13.2)	0.034
Lung	14 (20.3)	11.3 (3.26–30.4)	0.055
Peritoneum	21 (11.6)	5.30 (1.02–12.2)	0.269

IL-6 interleukin-6, IQR interquartile range, ECOG Eastern Cooperative Oncology Group, CA 19-9 carbohydrate antigen 19-9, ULN the upper limit of the normal range. The normal range is 0–35 U/mL. * In the population with metastatic disease.

**Table 2 biomedicines-12-00903-t002:** Effectiveness of first-line systemic chemotherapy in patients with advanced pancreatic cancer stratified by serum IL-6 levels.

Variables	Total (*n* = 65)	IL-6 High (*n* = 21)	IL-6 Low (*n* = 44)	*p* Value
First-line chemotherapy, *n* (%)				0.013
Gemcitabine-based	41 (63.1)	18 (85.7)	23 (52.3)	
Gemcitabine single	5 (7.7)	4 (19.0)	1 (2.3)	
Gemcitabine/Nab-paclitaxel	36 (55.4)	14 (66.7)	22 (50.0)	
FOLFIRINOX	24 (36.9)	3 (14.3)	21 (47.7)	
Best response, *n* (%)				
Partial response	11 (16.9)	2 (9.5)	9 (20.5)	
Stable disease	32 (49.2)	6 (28.6)	26 (59.1)	
Progressive disease	22 (33.9)	13 (61.9)	9 (20.4)	
Objective response rate, *n* (%)	11 (16.9)	2 (9.5)	9 (20.5)	0.480
Disease control rate, *n* (%)	43 (66.2)	8 (38.1)	35 (79.5)	0.001
Median PFS, months [95% CI]	5.9 [4.8–7.0]	2.1 [1.5–2.6]	9.1 [6.7–11.5]	<0.001
6-months PFS, % [95% CI]		10.7 [1.8–28.7]	64.9 [48.6–77.2]	

IL-6 interleukin-6, FOLFIRINOX fluorouracil, leucovorin, irinotecan, and oxaliplatin, PFS progression-free survival.

**Table 3 biomedicines-12-00903-t003:** Univariate and multivariate analyses of the clinical features and serum IL-6 levels for overall survival in patients with advanced pancreatic cancer.

	Overall Survival			
Variables	Univariate Analysis		Multivariate Analysis	
	HR (95% CI)	*p* Value	HR (95% CI)	*p* Value
Age ≥ 65 (vs. <65 year)	1.87 (1.13–3.10)	0.015	1.89 (1.02–3.47)	0.042
ECOG PS 2 (vs. ECOG PS 0–1)	2.78 (1.53–5.07)	0.001	1.93 (0.99–3.76)	0.053
Received chemotherapy (vs. none)	0.22 (0.11–0.42)	<0.001	0.29 (0.14–0.61)	0.001
Metastatic disease (vs. locally advanced)	1.72 (0.79–3.78)	0.175	1.80 (0.73–4.43)	0.199
IL-6 high (vs. low)	3.40 (2.04–5.65)	<0.001	2.31 (1.27–4.20)	0.006
NLR ≥ 3.5 (vs. <3.5)	2.94 (1.80–4.81)	<0.001	3.26 (1.77–6.00)	<0.001
CA 19-9 ≥ 59 × ULN (vs. <59 × ULN)	1.86 (1.11–3.13)	0.020	1.06 (0.58–1.94)	0.848

IL-6 interleukin-6, HR hazard ratio, CI confidence interval, ECOG PS Eastern Cooperative Oncology Group Performance Status, NLR neutrophil-to-lymphocyte ratio, CA 19-9 carbohydrate antigen 19-9, ULN the upper limit of normal range.

## Data Availability

All materials (data and images) reported in this article are available within the paper and its Appendix A.

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
