# Peer review of "The Role of Pretreatment Serum Interleukin 6 in Predicting Short-Term Mortality in Patients with Advanced Pancreatic Cancer"

_biomedicines, 2024, doi:10.3390/biomedicines12040903_

Round 1
Reviewer 1 Report
Comments and Suggestions for Authors
This manuscript intends to analyze interleukin 6 levels with disease severity and as potential prognostic indicator in patients with pancreatic ductal adenocarcinoma, also assessing how these biomarkers influence the efficacy of systemic chemotherapy.
There is no doubt about the impact of this study. However, the novelty of the work must be better demonstrated, because several relevant articles in this context were not referenced and discussed in the manuscript – examples:
https://www.ncbi.nlm.nih.gov/pmc/articles/PMC3670479/
https://www.frontiersin.org/journals/oncology/articles/10.3389/fonc.2022.964115/full
In addition it is important to refer two additional points:
-Ethics concerns – was the protocol analyzed by an Ethics Committee? This is determinant nowadays.
-The sample seems to be small – is this a representative sample? This must be demonstrated and explained in the manuscript
Comments on the Quality of English Language
Minor editing of English language required (e.g some long sentences)
Author Response
Reviewer 1
This manuscript intends to analyze interleukin 6 levels with disease severity and as potential prognostic indicator in patients with pancreatic ductal adenocarcinoma, also assessing how these biomarkers influence the efficacy of systemic chemotherapy.
Comment 1: There is no doubt about the impact of this study. However, the novelty of the work must be better demonstrated, because several relevant articles in this context were not referenced and discussed in the manuscript – examples:
https://www.ncbi.nlm.nih.gov/pmc/articles/PMC3670479/
https://www.frontiersin.org/journals/oncology/articles/10.3389/fonc.2022.964115/full
Response 1: Thank you for your insightful comments. As noted by the reviewer, there is existing research linking IL-6 levels with systemic treatment efficacy and survival outcomes. However, our study stands out because it not only involved a larger sample size but also integrated analyses of cytokines and established poor prognostic clinical features. Furthermore, our study's exploration of the relationship between the NLR—a significant prognostic marker in pancreatic cancer—and the proportions of neutrophils and lymphocytes, brings new insights. Following your detailed suggestions, we have included references to prior studies and extensively revised the discussion section to reflect our unique findings and outline future research directions
The manuscript is revised as follows:
In cohorts stratified by predetermined cutoff values for IL-6 and the IL-6:sIL-6Rα ratio, significant predictors of 6-month mortality, a discernible distinction in survival outcomes was observed correlating with these biomarkers. Elevated pretreatment levels of IL-6 were associated with accelerated disease progression and notably reduced survival durations, aligning with extant literature on metastatic PDAC that correlates increased serum IL-6 with adverse prognoses [25,26]. Moreover, the analysis focused solely on treated patients showed a correlation between high IL-6 levels and both poor OS and reduced PFS, suggesting a potential link between IL-6 and resistance to chemo-therapy. These findings align with prior research indicating that patients with elevated IL-6 levels experienced less favorable PFS outcomes when treated with gemcitabine monotherapy or FOLFIRINOX regimen [27,28]. Unlike previous studies, which either overlooked other clinical features or involved relatively small sample sizes, our study included a larger patient cohort and incorporated a variety of clinicopathological fac-tors in the survival analysis, making our findings more significant.
Regarding resistance to chemotherapy induced by IL-6, prior studies indicate that augmented IL-6 levels, which precipitate the activation of the JAK2-STAT3 pathway, are implicated in chemotherapy resistance [29,30]. Additionally, pro-inflammatory cytokines, including IL-6 within the PDAC tumor microenvironment (TME), are known to facilitate the evasion of anti-tumor immune responses and contribute to chemotherapy resistance [31]. Therefore, inhibiting the IL-6 pathway to suppress systemic inflammation or mitigate the impact of IL-6 on tumor and immune cells within the TME remains an attractive potential therapeutic target.
Regarding absolute white blood cell counts, individuals with elevated IL-6 levels exhibited notably higher absolute neutrophil counts, while absolute lymphocyte counts were maintained at a relatively constant level between the two groups. Furthermore, no substantial differences were observed in the proportions of CD4+ and CD8+ lymphocytes between the groups. These findings indicate that the adverse prognosis linked to a higher NLR is more significantly influenced by severe systemic inflammation from increased neutrophils, rather than by the immunosuppressive effects stemming from a reduction in lymphocytes. Therefore, treatment strategies for pancreatic cancer that focus on reducing systemic inflammation and preventing neutrophil infiltration into the tumor might be more beneficial than targeting immunosuppressive mechanisms alone.
Comment 2: In addition, it is important to refer two additional points:
-Ethics concerns – was the protocol analyzed by an Ethics Committee? This is determinant nowadays.
Response 2: Thank you for your supportive comments. Our study was reviewed and approved by the Institutional Review Board (IRB) of The Catholic University of Korea, Seoul St. Mary’s Hospital. The protocol and all procedures involving subjects were conducted with IRB approval. We ensured that all participants were fully informed about the purpose and process of the research, confirmed their understanding, and obtained informed consent before blood sampling. This information is noted in the manuscript as follows.
Institutional Review Board Statement: All procedures were performed in accordance with Ko-rean regulations and the Declaration of Helsinki. This study was approved by the Institutional Review Board (IRB) of The Catholic University of Korea, Seoul St. Mary’s Hospital (approval ID: KC23SASI0863).
Informed Consent Statement: Written informed consent has been obtained from the patients to publish this paper before blood collection.
Comment 3: The sample seems to be small – is this a representative sample? This must be demonstrated and explained in the manuscript
Response 3: We sincerely appreciate the reviewer for their perceptive observations. Indeed, as noted, the sample size of our study is modest, and we concur that it may not sufficiently represent all patients with advanced pancreatic cancer. However, it is important to consider that pancreatic cancer, unlike more prevalent cancers such as breast, lung, and colon, has a lower incidence and notably poorer prognosis, which inherently restricts the number of available patients for treatment post-diagnosis. Given these unique aspects of pancreatic cancer, analyzing a cohort of 77 patients, 65 of whom underwent systemic chemotherapy, remains significant. Nonetheless, we acknowledge that the overall sample size does not provide robust evidence; accordingly, we have included a discussion of these limitations in our manuscript. We are committed to conducting further exploratory analyses in future larger-scale clinical trials. Thank you for your invaluable feedback.
The manuscript is revised as follows:
Several limitations of our study should be noted. Firstly, this research was con-ducted as a retrospective study at a single institution and involved a relatively small sample size, both of which could act as confounding factors. Secondly, the relationship between treatment efficacy and IL-6 levels was not thoroughly assessed, as measurements of IL-6 were not taken at multiple points after treatment. Additionally, IL-6 levels can increase due to bacterial infections, which raises the possibility that elevated mortality might have resulted from infections rather than tumor progression.
Comment 4: Minor editing of English language required (e.g some long sentences)
Response 4: We truly appreciate your thorough review. As the reviewer pointed out, we have revisited and revised the lengthy and complex sentences to enhance readability and clarity. Additionally, we have edited a few minor aspects of the manuscript, such as the omission of descriptions for some abbreviations.
The manuscript is revised as follows:
Interleukin-6 (IL-6), a pro-inflammatory cytokine, facilitates cancer development by activating the Janus kinase 2 (JAK2)-signal transducer and activator of transcription 3 (STAT3) pathways [12]. This leads to enhanced tumor vascularization, which in turn promotes the migration and metastasis of cancer cells [13].
To address this, we employed receiver operating characteristic (ROC) curve analysis to define an optimal prognostic cutoff value for IL-6 and the IL-6:sIL-6Rα ratio in predicting survival six months after enrollment. This allowed us to classify patients into two cohorts—those with high versus low levels of IL-6 and the IL-6:sIL-6Rα ratio—based on the ideal cutoff value.
OS was defined as the duration in months from the advanced disease diagnosis until death from any cause or the last follow-up. Progression-free survival (PFS) was defined as the time between the start of chemo-therapy and either documented disease progression or death.
The median age was 67 years, with the majority (80.5%) having an Eastern Cooperative Oncology Group (ECOG) score of 0–1.
This inefficacy may stem from the higher concentrations of sIL-6R compared to IL-6 it-self, hinting that targeting sIL-6R might not yield significant benefits. Moreover, the rapid formation of a heterodimer complex between plasma-secreted IL-6, sIL-6R, and sgp130 raises possibility of therapeutic targeting. Using an antibody like sgp130Fc protein to target the IL-6/sIL-6R complex could offer a therapeutic advantage [21].
Reviewer 2 Report
Comments and Suggestions for Authors
In this work the role of pretreatment serum interleukin 6 in predicting short-term mortality in patients with advanced pancreatic cancer is described. The authors studied the correlation between pretreatment serum IL-6 levels, encompassing proinflammatory cytokine levels, and survival outcomes in advanced pancreatic cancer patients, while also assessing how these biomarkers influence the efficacy of systemic chemotherapy. In addition, the correlation between IL-6 and the systemic inflammatory marker, NLR, by analyzing the proportion and absolute count of neutrophils and lymphocytes according to IL-6 levels was examined. Thus, the manuscript looks like an Article and may be published after minor revision.
Notes:
1. The significance of this work should be reported more detailed in the Introduction. The role of Interleukin-6 as a biomarker for the selection of treatment strategies for pancreatic cancer disease should be noted more detailed. Are such examples known in the literature?
2. What do the red and blue dotted lines mean in Figure 2? It should be mentioned.
3. Promising application of new obtained results should be mentioned in conclusion. The conclusions of the work should be written more detailed. What correlations were found? How do biomarkers influence the effectiveness of systemic chemotherapy?
Author Response
Reviewer 2
In this work the role of pretreatment serum interleukin 6 in predicting short-term mortality in patients with advanced pancreatic cancer is described. The authors studied the correlation between pretreatment serum IL-6 levels, encompassing proinflammatory cytokine levels, and survival outcomes in advanced pancreatic cancer patients, while also assessing how these biomarkers influence the efficacy of systemic chemotherapy. In addition, the correlation between IL-6 and the systemic inflammatory marker, NLR, by analyzing the proportion and absolute count of neutrophils and lymphocytes according to IL-6 levels was examined. Thus, the manuscript looks like an Article and may be published after minor revision.
Comment 1: The significance of this work should be reported more detailed in the Introduction. The role of Interleukin-6 as a biomarker for the selection of treatment strategies for pancreatic cancer disease should be noted more detailed. Are such examples known in the literature?
Response 1: Thank you for your invaluable feedback. We agree that there needs to be a more detailed explanation of what pre-treatment IL-6 levels signify in patients with pancreatic cancer and why this warrants investigation. IL-6 is associated with poor prognosis in various cancers, including pancreatic cancer, where it also appears to be linked to resistance to systemic chemotherapy. However, the mechanisms underlying these relationships have not been clearly identified. Additionally, in pancreatic cancer, excessive systemic inflammation is widely recognized as being associated with poor outcomes. Therefore, understanding the role of IL-6 in this process is crucial. We aim to investigate this further and explore the potential for new therapeutic strategies that could be applied to the treatment of pancreatic cancer through additional research.
The manuscript is revised as follows:
Interleukin-6 (IL-6), a pro-inflammatory cytokine, facilitates cancer development by activating the Janus kinase 2 (JAK2)-signal transducer and activator of transcription 3 (STAT3) pathways [12]. This leads to enhanced tumor vascularization, which in turn promotes the migration and metastasis of cancer cells [13]. Additionally, in patients with advanced PDAC, elevated pretreatment levels of IL-6 are associated with poor survival outcomes and resistance to chemotherapy [12].
However, the relationship between serum proinflammatory cytokines and the effectiveness of systemic chemotherapy, along with survival outcomes in patients with advanced pancreatic cancer, remains largely unexplored. Investigating the association be-tween pretreatment proinflammatory cytokines and short-term mortality or chemo-therapy resistance in these patients could prevent unnecessary treatment in those predicted not to respond. Additionally, elucidating the relationship between IL-6 and systemic inflammatory biomarkers may help characterize the features of pancreatic cancer that are commonly known to be unresponsive to immunotherapy, potentially leading to new therapeutic strategies.
Comment 2: What do the red and blue dotted lines mean in Figure 2? It should be mentioned.
Response 2: We are truly grateful for your thorough review. The dot markings on the survival curve in Figure 2 represent the addition of 95% CI values at each point. In the previous Figure 1, the overall survival curve was presented without these markings, as the 95% CI values are not essential for survival curves. Therefore, we have created a new graph without these values and attached it.
The manuscript is revised as follows:
Comment 3: Promising application of new obtained results should be mentioned in conclusion. The conclusions of the work should be written more detailed. What correlations were found? How do biomarkers influence the effectiveness of systemic chemotherapy?
Response 3: We are truly grateful for your thorough review. We fully agree with the suggestion that our study results require more detailed analysis and discussion on future directions. A key finding of our research is the association of serum IL-6 levels and the adjusted IL-6:sIL-6Rα ratio with poor survival outcomes and resistance to systemic chemotherapy. Additionally, we have conducted a more detailed analysis of the relationship with the NLR, which is increasingly recognized for its prognostic significance. Our findings suggest that high IL-6 levels and exacerbated systemic inflammation due to an increase in neutrophils might be linked to treatment resistance and poor survival outcomes in pancreatic cancer. Further research is certainly needed to explore the role of IL-6 in the TME and its associated effects on tumor-infiltrating neutrophils and the activation of the IL-6 pathway in the TME. We have made extensive revisions related to this discussion in the discussion section, and the conclusions have also been revised accordingly.
The manuscript is revised as follows:
In cohorts stratified by predetermined cutoff values for IL-6 and the IL-6:sIL-6Rα ratio, significant predictors of 6-month mortality, a discernible distinction in survival outcomes was observed correlating with these biomarkers. Elevated pretreatment levels of IL-6 were associated with accelerated disease progression and notably reduced survival durations, aligning with extant literature on metastatic PDAC that correlates increased serum IL-6 with adverse prognoses [25,26]. Moreover, the analysis focused solely on treated patients showed a correlation between high IL-6 levels and both poor OS and reduced PFS, suggesting a potential link between IL-6 and resistance to chemo-therapy. These findings align with prior research indicating that patients with elevated IL-6 levels experienced less favorable PFS outcomes when treated with gemcitabine monotherapy or FOLFIRINOX regimen [27,28]. Unlike previous studies, which either overlooked other clinical features or involved relatively small sample sizes, our study included a larger patient cohort and incorporated a variety of clinicopathological fac-tors in the survival analysis, making our findings more significant.
Regarding resistance to chemotherapy induced by IL-6, prior studies indicate that augmented IL-6 levels, which precipitate the activation of the JAK2-STAT3 pathway, are implicated in chemotherapy resistance [29,30]. Additionally, pro-inflammatory cytokines, including IL-6 within the PDAC tumor microenvironment (TME), are known to facilitate the evasion of anti-tumor immune responses and contribute to chemotherapy resistance [31]. Therefore, inhibiting the IL-6 pathway to suppress systemic inflammation or mitigate the impact of IL-6 on tumor and immune cells within the TME remains an attractive potential therapeutic target.
Regarding absolute white blood cell counts, individuals with elevated IL-6 levels exhibited notably higher absolute neutrophil counts, while absolute lymphocyte counts were maintained at a relatively constant level between the two groups. Furthermore, no substantial differences were observed in the proportions of CD4+ and CD8+ lymphocytes between the groups. These findings indicate that the adverse prognosis linked to a higher NLR is more significantly influenced by severe systemic inflammation from increased neutrophils, rather than by the immunosuppressive effects stemming from a reduction in lymphocytes. Therefore, treatment strategies for pancreatic cancer that focus on reducing systemic inflammation and preventing neutrophil infiltration into the tumor might be more beneficial than targeting immunosuppressive mechanisms alone.
In conclusion, pretreatment IL-6 levels and the IL-6:sIL-6Rα ratio were associated with worse survival outcomes and chemotherapy resistance, with this association consistent in multivariate analysis. These insights could enhance the management of patients with advanced PDAC by identifying those most likely to benefit from systemic treatment. IL-6 was closely linked to exacerbated systemic inflammation, primarily driven by an increase in neutrophils. Further research is essential to explore how circulating IL-6 levels relate to its presence in the TME, understand its immunosuppressive role, and examine the correlation between increased circulating and tumor-infiltrating neutrophils in advanced pancreatic cancer.
Round 2
Reviewer 1 Report
Comments and Suggestions for Authors
The document was improved and now it is more acceptable for publication.